# Endogenous IL-33 Accelerates Metacestode Growth during Late-Stage Alveolar Echinococcosis

Brice Autier,[a] Christelle Manuel,[b] Britta Lundstroem-Stadelmann,[c] Jean-Philippe Girard,[d] Bruno Gottstein,[e] Jean-Pierre Gangneux,[a] Michel Samson,[b] Florence Robert-Gangneux,[a] Sarah Dion[b]

aIRSET (UMR_S 1085), INSERM (Institut de recherche en santé, environnement et travail), EHESP, CHU Rennes, University of Rennes, Rennes, France
bIRSET (UMR_S 1085), INSERM (Institut de recherche en santé, environnement et travail), EHESP, University of Rennes, Rennes, France
cInstitute of Parasitology, Department of Infectious Diseases and Pathobiology, Vetsuisse Faculty, University of Bern, Bern, Switzerland
dInstitut de Pharmacologie et de Biologie Structurale (IPBS), Université de Toulouse, Toulouse, France
eInstitute of Infectious Diseases, Faculty of Medicine, University of Bern, Bern, Switzerland

Florence Robert-Gangneux and Sarah Dion contributed equally to this article. Author order was determined based on involvement in conceptualization.

**ABSTRACT** During the course of the infectious disease alveolar echinococcosis (AE), the larval stage of *Echinococcus multilocularis* develops in the liver, where an initial Th1/Th17 immune response may allow its elimination in resistant individuals. In patients susceptible to infection and disease, the Th2 response initiates later, inducing tolerance to the parasite. The role of interleukin 33 (IL-33), an alarmin released during necrosis and known to drive a Th2 immune response, has not yet been described during AE. Wild-type (WT) and IL-33$^{-/-}$ C57BL/6J mice were infected by peritoneal inoculation with *E. multilocularis* metacestodes and euthanized 4 months later, and their immune response were analyzed. Immunofluorescence staining and IL-33 enzyme-linked immunosorbent assay (ELISA) were also performed on liver samples from human patients with AE. Overall, metacestode lesions were smaller in IL-33$^{-/-}$ mice than in WT mice. IL-33 was detected in periparasitic tissues, but not in mouse or human serum. In infected mice, endogenous IL-33 modified peritoneal macrophage polarization and cytokine profiles. Th2 cytokine concentrations were positively correlated with parasite mass in WT mice, but not in IL-33$^{-/-}$ mice. In human AE patients, IL-33 concentrations were higher in parasitic tissues than in distant liver parenchyma. The main sources of IL-33 were CD31$^+$ endothelial cells of the neovasculature, present within lymphoid periparasitic infiltrates together with FOXP3$^+$ T$_{reg}$s. In the murine model, periparasitic IL-33 correlated with accelerated parasite growth putatively through the polarization of M2-like macrophages and release of immunosuppressive cytokines IL-10 and transforming growth factor $\beta$1 (TGF-$\beta$1). We concluded that IL-33 is a key alarmin in AE that contributes to the tolerogenic effect of systemic Th2 cytokines.

**IMPORTANCE** Infection with the metacestode stage of *Echinococcus multilocularis*, known as alveolar echinococcosis, is the most severe cestodosis worldwide. However, less than 1% of exposed individuals, in which the immune system is unable to control the parasite, develop the disease. The factors responsible for this interindividual variability are not fully understood. In this *in vivo* study comparing wild-type and IL-33$^{-/-}$ infected mice, together with data from human clinical samples, we determined that IL-33, an alarmin released following tissue injury and involved in the pathogenesis of cancer and asthma, accelerates the progression of the disease by modulating the periparasitic microenvironment. This suggests that targeting IL-33 could be of interest for the management of patients with AE, and that IL-33 polymorphisms could be responsible for increased susceptibility to AE.

**KEYWORDS** IL-33, *Echinococcus multilocularis*, alveolar echinococcosis, angiogenesis, ST2, IL-1RL1

Address correspondence to Sarah Dion, sarah.dion@univ-rennes1.fr.

The authors declare no conflict of interest.

Alveolar echinococcosis (AE) is a widespread zoonosis caused by the parasite *Echinococcus multilocularis*, a cestode infecting the intestine of wild foxes and a wide range of canids. The ingestion of parasite eggs, spread in the environment through fox droppings, and possibly contaminated food and water leads to AE, one of the most lethal and severe helminthic diseases in humans (1). After its translocation through the digestive tract, the parasite mainly infects the liver. There, the parasite forms a heterogeneous mass consisting of parasitic microcysts, called metacestodes (2), host connective tissue, immune cells, and blood vessels. In susceptible hosts, the larval metacestode continues to grow indefinitely until the death of the host. Furthermore, it is capable of spreading via metastases, thereby forming secondary lesions in other organs (3). Due to this clinical presentation, the typical imaging, and the risk of recurrence, AE is often compared to cancer (4).

During the early stages of infection, the parasite triggers the recruitment of myeloid cells, such as polymorphonuclear cells, monocytes, and dendritic cells, which initiate a pro-inflammatory Th1/Th17 adaptative response (5, 6). In most human infections (99% of cases, as shown in Central European countries), this pro-inflammatory response is able to control the parasite's development, leading to parasite clearance or abortion of established lesions (7). However, in less than 1% of cases, immunity then switches to a mixed Th1/Th2 phenotype and expands its $T_{reg}$ component, allowing parasite survival and accelerating metacestode growth (8, 9).

Interleukin 33 (IL-33) is a nuclear cytokine of the IL-1 family which is released upon cellular damage (10). It interacts with the receptor ST2 (also called IL-1RL1), naturally expressed by tissue-resident immune cells such as mast cells, type-2 innate lymphoid cells (ILC2), and $T_{reg}$ lymphocytes (10), as well as by circulating immune cells such as Th2 cells and eosinophils (11). However, ST2 expression can be induced in many cell types, including neutrophils and macrophages (10). IL-33 is mostly described to have "tolerogenic" effects because it can induce Th2 immunity; it can also regulate pro-inflammatory reaction through $T_{reg}$ activation and polarize macrophages into an "M2" phenotype (12–14). However, the roles of IL-33 are pleiotropic and depend on the source and location of inflammation and the presence/absence of a systemic inflammation (15). In infectious diseases, IL-33 can be either deleterious, as it induces inflammation and fibrosis against helminths (16, 17) and tolerance against intracellular parasites such as *Leishmania* (18, 19); or beneficial, as it has anti-inflammatory properties and protective effects against intestinal helminths (20–22).

This study aimed to describe the role of IL-33 in AE using an *in vivo* model of peritoneal infection. Because IL-33 has been shown to promote Th2 and $T_{reg}$ activity, we hypothesized that it could be involved in late-stage tolerance and/or anergy. Therefore, we studied a 4-month course of infection in wild-type (WT) and IL-33$^{-/-}$ mice. We showed that IL-33 favors metacestode development in mice. These findings are consistent with a local role of IL-33, as ST2$^+$ cell populations are recruited at the site of infection (T CD4$^+$ cells, macrophages, neutrophils, eosinophils) and IL-33 is detected in periparasitic tissues, but not in sera. We also showed in hepatic biopsy samples from human patients with AE that IL-33 is expressed in periparasitic neovasculature tissue, suggesting that parasite growth could be associated with IL-33 release upon endothelial damage.

## RESULTS

**Endogenous IL-33 favors metacestode growth *in vivo*.** To evaluate the role of IL-33 in AE development, *E. multilocularis*-infected IL-33$^{-/-}$ mice and control WT littermates were analyzed after 4 months postinfection (pi). At this late stage of infection, the peritoneal metacestodes which developed in the WT mice had a higher weight (12.1 $\pm$ 5.6 g) compared to those that developed in the IL-33$^{-/-}$ mice (7.7 $\pm$ 3.5 g, $P \leq 0.05$) (Fig. 1A). This higher growth rate was confirmed by the weight of the whole mice, which was significantly higher in the WT group compared to the IL-33$^{-/-}$ group, at 14 and 18 weeks pi. ($P \leq 0.01$ and $P \leq 0.001$, respectively) (Fig. 1B). Expression of the parasitic *Em14-3-3* gene, a metacestode viability marker, was reduced in the IL-33$^{-/-}$ group ($P \leq 0.01$) (Fig. 1C). Hepatic secondary lesions were observed in 16.7% (2/12) and 13.3% (2/15) of mice in the WT and IL-33$^{-/-}$ groups, respectively ($P = 1.0$, Fisher's exact test). Metacestodes took the shape of heterogeneous

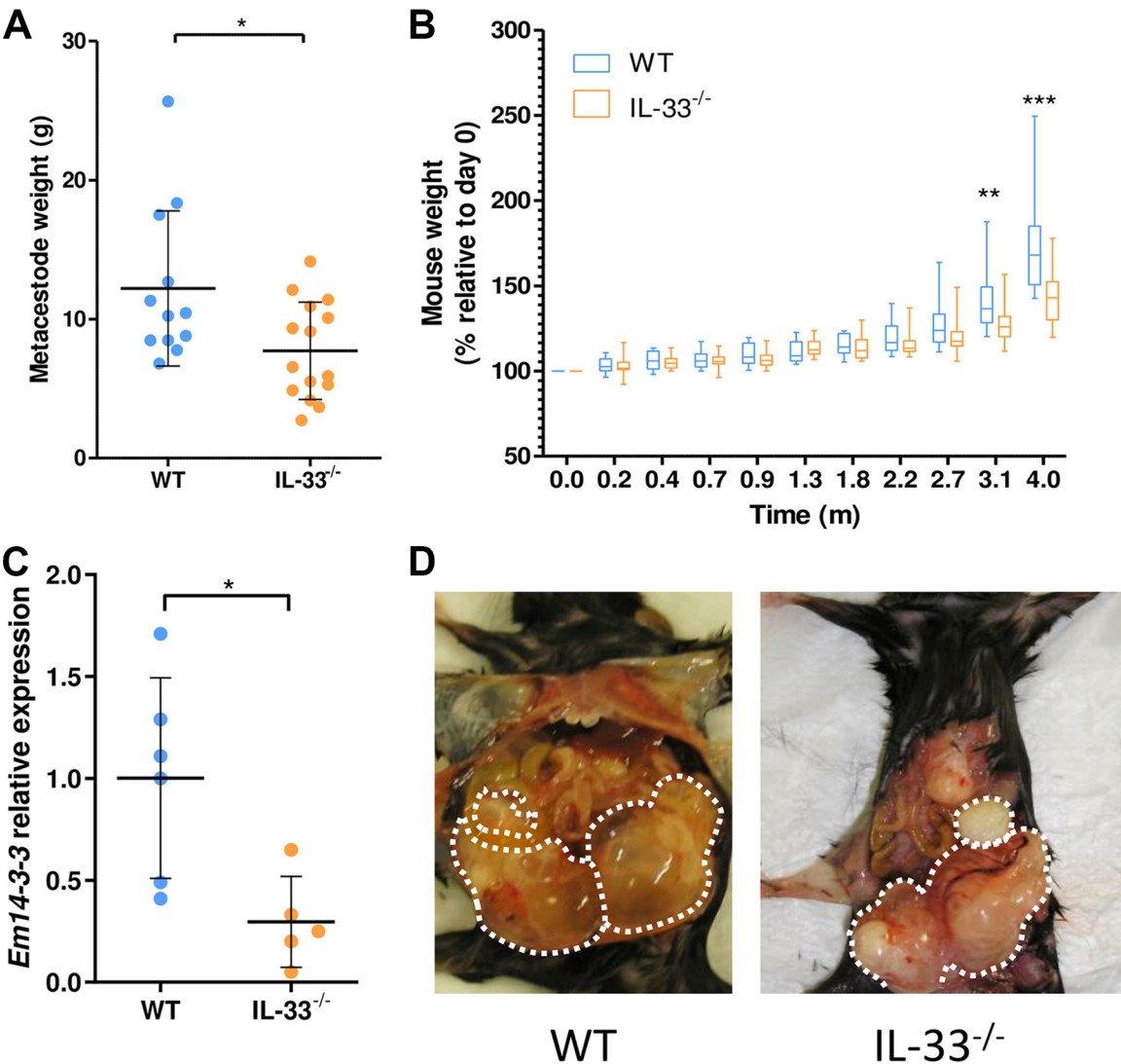

**FIG 1** Endogenous interleukin 33 (IL-33) is associated with accelerated course of alveolar echinococcosis (AE). (A) Wet weight of peritoneal metacestodes collected from mice after 4 months of infection. (B) Mouse weight evolution during the course of infection. (C) Quantification of $Em14$-$3$-$3$ gene expression, a parasite viability marker, normalized using the threshold cycle ($2^{-\Delta\Delta CT}$) method and $Echinococcus\ multilocularis$ ActII gene. (D) Representative pictures of peritoneal lesions (surrounded by dotted lines) from wild-type (WT) and IL-33$^{-/-}$ mice. Panels A and B: pooled data from 2 independent experiments, with 12 WT and 14 IL-33$^{-/-}$ mice. Panel C: data obtained from one experiment, with 6 WT and 5 IL-33$^{-/-}$ mice. Plots show mean $\pm$ standard deviation (SD). Significance was tested using Mann-Whitney U test (A and C) and two-way analysis of variance (ANOVA) for repeated measures with Bonferroni's post-tests (B). *, $P \leq 0.05$; **, $P \leq 0.01$; ***, $P \leq 0.001$.

microcystic structures intermingled with host connective tissues in both groups (Fig. 1D). IL-33 was not detectable in mouse sera at 4 months pi but was detected in tissues surrounding the metacestodes of WT mice (406.2 $\pm$ 185 ng/g of protein, $n = 5$, supplemental file 1).

**Endogenous IL-33 induces a tolerogenic phenotype in the periparasitic microenvironment.** Immune populations in the peritoneum were characterized by multiparametric flow cytometry. Peritoneal infection with $E.\ multilocularis$ led to significant recruitment of monocyte/macrophages ($P \leq 0.001$ and $P \leq 0.05$ for WT and IL-33$^{-/-}$ mice, respectively), other myeloid cells ($P \leq 0.01$), and natural killer (NK) cells ($P \leq 0.01$), together with a concomitant decrease in the resident B cells ($P \leq 0.01$), equally in both the WT and IL-33$^{-/-}$ groups, compared to uninfected mice (Fig. 2A). The proportions of other cell populations were not significantly different between groups. The infiltrate of myeloid cells was mainly composed of monocytes/macrophages, neutrophils, and, to a lesser extent, eosinophils and dendritic cells, without significant differences between WT and IL-33$^{-/-}$ mice (supplemental file 1). The proportion of CD206$^+$ monocytes/macrophages was significantly decreased in IL-33$^{-/-}$ mice (Fig. 2B, $P \leq 0.05$), together with the expression of M2 markers such as the

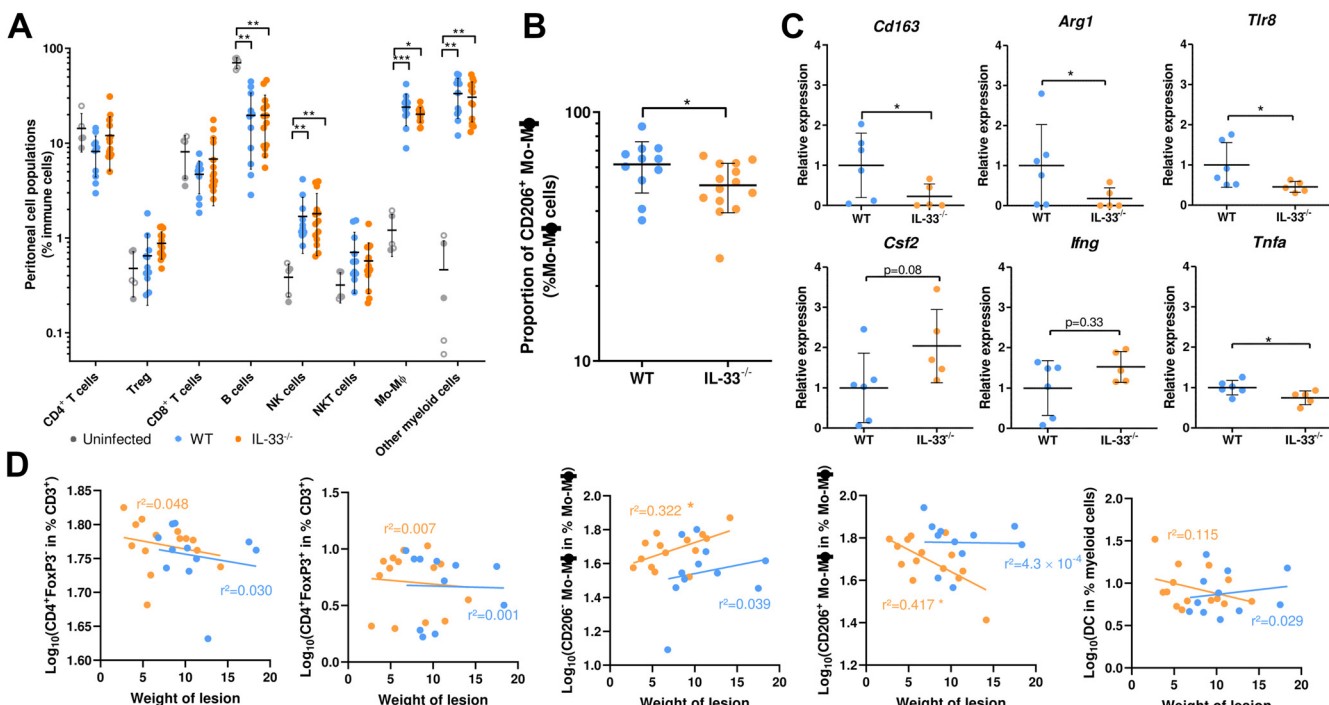

**FIG 2** Endogenous IL-33 modulates peritoneal cell populations during AE. (A) Composition of peritoneal cell infiltrates at 4 months postinfection. (B) Proportion of CD206$^+$ monocytes/macrophages in peritoneal infiltrates. (C) Quantification of gene expression in the periparasitic tissues, normalized using 2$^{-\Delta\Delta CT}$ method and the *Gapdh* and *Ppid* endogenous controls. (D) Linear regression between the weight of lesions and proportions of peritoneal cell populations of WT (blue dots and line) and IL-33$^{-/-}$ mice (orange dots and line). Panels A, B, and D: pooled data from 2 independent experiments, with 5 uninfected (2 WT [close circles] and 3 IL-33$^{-/-}$ [open circles] mice) and 11 infected WT and 14 IL-33$^{-/-}$ mice. Panel C: data obtained from one experiment with 6 WT and 5 IL-33$^{-/-}$ mice. Plots show means ± SD (A, B, and C) and the $r^2$ correlation coefficient (D). Significance was tested using the Kruskal-Wallis test with Dunn's multiple-comparison *post hoc* analysis (A), the Mann-Whitney U test (B and C) and the F test (D). *, $P \leq 0.05$; **, $P \leq 0.01$; ***, $P \leq 0.001$.

*Cd163*, *Arg1*, and *Tlr8* genes (Fig. 2C, $P \leq 0.05$), suggesting decreased M2 macrophage polarization in these mice. Expression of M1 markers (*Csf2*, *Ifng*) was not affected by IL-33 knockout, except for *Tnfa* which was, although significantly ($P \leq 0.05$), moderately repressed. Lesion weight was not correlated with the proportions of immune cells, except for the proportions of CD206$^-$ and CD206$^+$ macrophages in IL-33$^{-/-}$ mice ($P \leq 0.05$), which increased and decreased, respectively, with lesion weight (Fig. 2D).

The dosages of cytokines in peritoneal lavages, which are related to local immune events, showed higher IL-10 concentrations in infected WT mice compared to uninfected ($P \leq 0.05$) and infected IL-33$^{-/-}$ animals ($P \leq 0.05$, Fig. 3). Concentrations of TGF-$\beta$1 ($P \leq 0.01$), tumor necrosis factor $\alpha$ (TNF-$\alpha$; $P \leq 0.01$), IL-2 ($P \leq 0.01$), IL-9 ($P \leq 0.05$), and chemokine ligand 1 (CXCL1; $P \leq 0.05$) were significantly increased in infected WT, but not in infected IL-33$^{-/-}$ mice; conversely, CCL22 was significantly increased in IL-33$^{-/-}$ mice but not in WT mice ($P \leq 0.05$). Compared to that in uninfected mice, other cytokines were either unchanged (IL-17A, interferon $\gamma$ [IFN-$\gamma$]), equally increased in both WT and IL-33$^{-/-}$ mice (IL-4, IL-5, IL-6), or not detected (IL-17F and IL-13). Altogether, these data suggest that IL-33 is responsible for local macrophage polarization and mixed Th1/Th2 cytokine profile, including secretion of the anti-inflammatory cytokine IL-10.

**Endogenous IL-33 has little influence on hepatic immune response during peritoneal infection.** Flow cytometry analysis of the liver-infiltrating cells showed significant recruitment of myeloid cells upon peritoneal infection with *E. multilocularis* ($P \leq 0.001$ and $P \leq 0.05$ for WT and IL-33$^{-/-}$ mice, respectively), together with a decrease in the NK T cell population ($P \leq 0.01$ for both WT and IL-33$^{-/-}$ mice), compared to that in uninfected mice (Fig. 4A). The proportions of other analyzed immune cell populations were not altered by the adjacent (peritoneal) infection. The CD4$^+$:CD8$^+$ T cell ratio was not modified by IL-33 knockout (Fig. 4B). The myeloid infiltrate was mainly composed of neutrophils and CD206$^-$ macrophages, and distributions of myeloid populations were not different between WT and

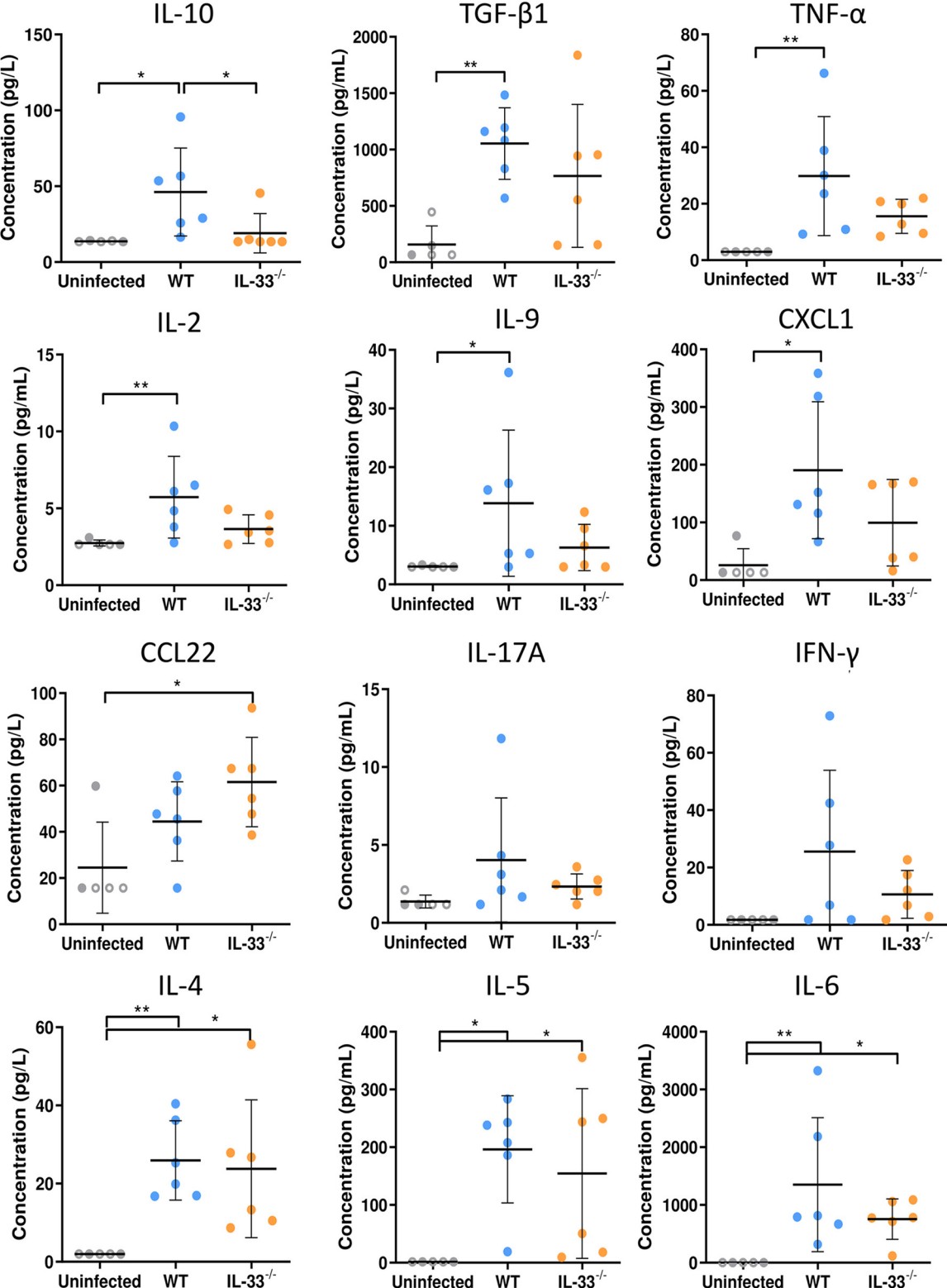

**FIG 3** Endogenous IL-33 modulates concentrations of cytokines in peritoneal lavages during AE. Cytokine concentrations determined by flow cytometry in peritoneal lavages of WT and IL-33$^{-/-}$ mice at 4 months postinfection. Data obtained from one experiment, with 5 uninfected (2 WT [close circles] and 3 IL-33$^{-/-}$ [open circles]) and 6 infected WT and 6 infected IL-33$^{-/-}$ mice. Plots show means ± SD. Significance was tested using the Kruskal-Wallis test with Dunn's multiple-comparison *post hoc* analysis. *, $P \leq 0.05$; **, $P \leq 0.01$; ***, $P \leq 0.001$.

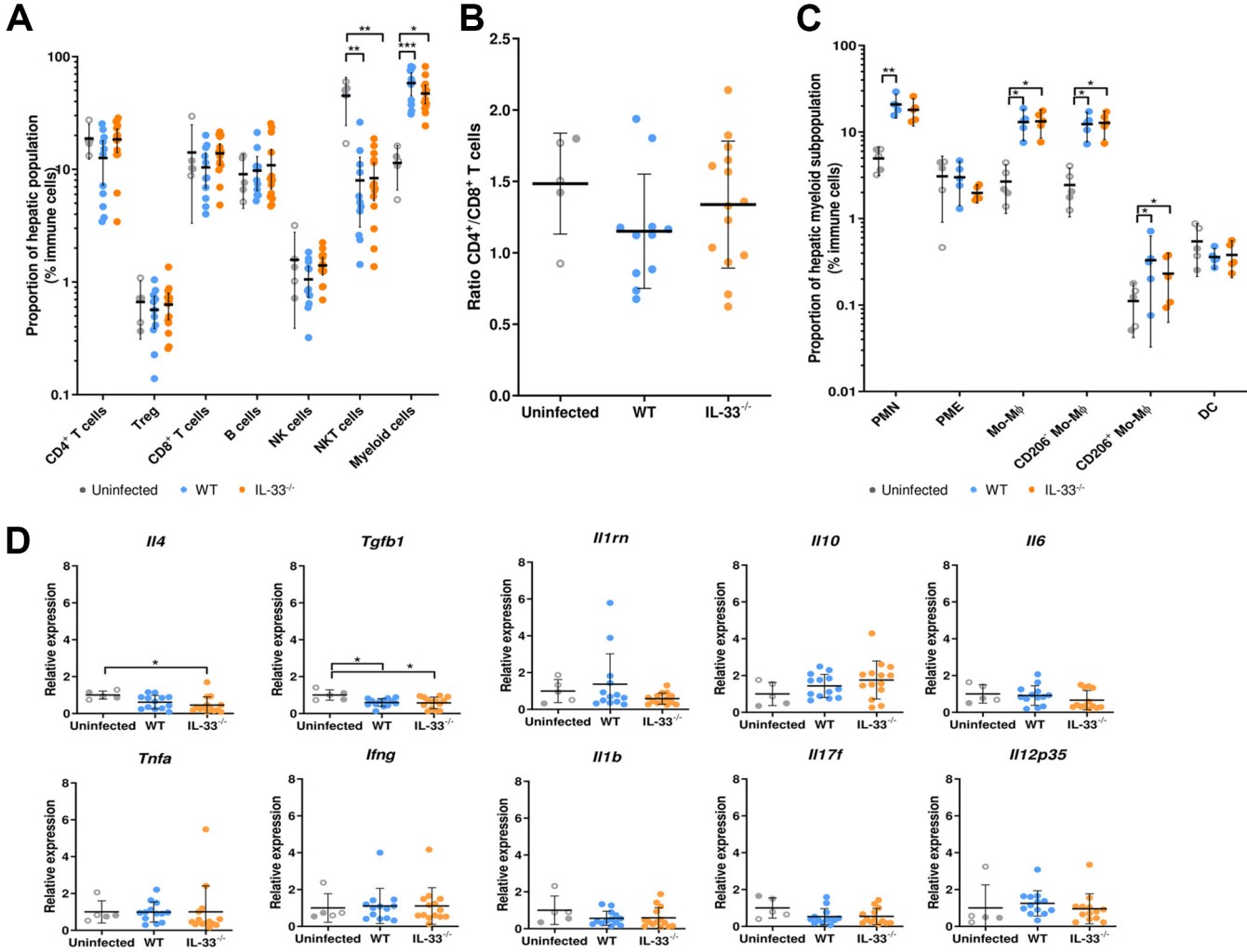

**FIG 4** Endogenous IL-33 has little impact on cellular and transcription profiles in the liver. (A) Composition of liver cellular infiltrates at 4 months postinfection. Gating strategy was the same as used for peritoneal cells, except that Panel 2 was applied in 1 on 2 experiments. (B) Ratio of CD4$^+$ to CD8$^+$ T cells. (C) Myeloid subpopulations of liver cellular infiltrates. (D) Quantification of gene expression in the liver, normalized using the $2^{-\Delta\Delta CT}$ method and the *Gapdh* and *Actb* endogenous controls. Panels A, B, and D: pooled data from 2 independent experiments, with 5 uninfected (2 WT [close circles] and 3 IL-33$^{-/-}$ [open circles]) and 11 infected WT and 14 infected IL-33$^{-/-}$ mice. Panel C: data obtained from one experiment, with 5 mice in each group. Plots show means ± SD. Significance was tested using the Kruskal-Wallis test with Dunn's multiple-comparison *post hoc* analysis. *, $P \leq 0.05$; **, $P \leq 0.01$; ***, $P \leq 0.001$.

IL-33$^{-/-}$ mice (Fig. 4C). A moderate but statistically significant repression of *Il4* expression was observed in IL-33$^{-/-}$ mice (0.45 ± 0.45) compared to the uninfected controls (1 ± 0.22, $P \leq 0.05$) (Fig. 4D). Quantification of mRNA expression in the liver did not show differences between WT and IL-33$^{-/-}$ mice for other genes, including those for Th1 (*Tnfa*, *Il12p35*, *Ifng*, *Il1b*), Th2 (*Il6*, *Tgfb1*, *Il10*), Th17 (*Il17f*), or anti-inflammatory (*Il1rn*) mediators. *Tgfb1* was repressed in both WT and IL-33$^{-/-}$ mice compared to uninfected controls ($P \leq 0.05$).

**Systemic response requires endogenous IL-33 to enhance metacestode growth.** Overall, the mean serum concentrations of all detected cytokines (IFN-$\gamma$, IL-17A, IL-17F, IL-5, IL-4, IL-6, IL-9, IL-1$\beta$, and TNF-$\alpha$) except for IL-13, IL-12p40, and IL-23, significantly increased over time in both groups of mice (Fig. 5A and Fig. S2). Serum concentrations of IL-5 at 1 month pi (early stage) were higher in WT mice than in IL-33$^{-/-}$ mice ($P \leq 0.05$, Fig. 5A). At 2 months pi (intermediate stage), serum concentrations of IL-12p40 were higher in WT mice than in IL-33$^{-/-}$ mice ($P \leq 0.05$, Fig. S2). At a later stage, higher IL-17F concentrations were observed in the sera of IL-33$^{-/-}$ mice ($P \leq 0.05$ at 4 months), compared to WT mice, suggesting that IL-33 could be associated with a decreased Th17 response (Fig. 5A). No differences were observed between groups for other cytokines. Comparison of cytokine

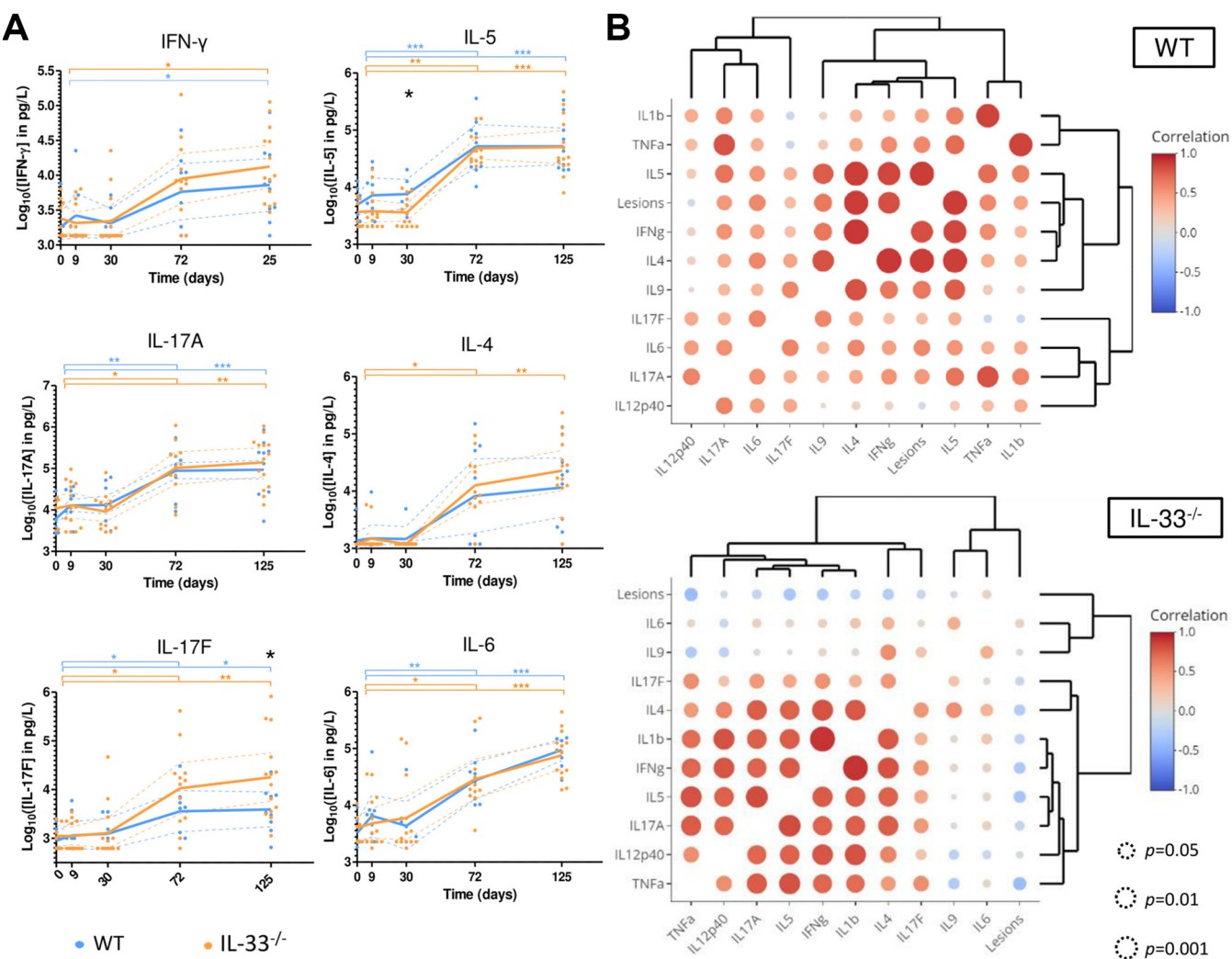

**FIG 5** Endogenous IL-33 modulates systemic cytokine profiles during AE. (A) Cytokines concentrations determined by flow cytometry in the serum of WT and IL-33$^{-/-}$ mice during the course of infection. Blue and orange symbols indicate statistical significance over time within each group of mice (Kruskal-Wallis with *post hoc* Dunn's multiple-comparison test), and black symbols stand for comparison between mice groups at a given time point (Mann-Whitney U test). Concentrations were normalized by logarithmic transformation. Dotted lines show 95% confidence intervals. (B) Heatmaps of pairwise correlation coefficients analysis between serum cytokines concentrations at 4 months postinfection. Dot color and size indicate correlation coefficients and their statistical significance, respectively. Pooled data from 2 independent experiments, with 10 WT and 14 IL-33$^{-/-}$ mice. The parameter "lesions" stands for the weight of the metacestodes 4 months postinfection. *, $P \leq 0.05$; **, $P \leq 0.01$; ***, $P \leq 0.001$.

concentrations in peritoneal lavages showed that IL-17F and IL-13 were detected only in sera, whereas IL-10 was detected only in peritoneal lavages (Fig. 3).

Analysis of pairwise correlation coefficients at 4 months pi showed that in the WT group, the weight of parasitic lesions was significantly and positively correlated with Th2 cytokines (Fig. 5B), such as IL-4 ($r = 0.86$, $P \leq 0.001$) and IL-5 ($r = 0.88$, $P \leq 0.001$), but also with the Th1 IFN-$\gamma$ ($r = 0.81$, $P \leq 0.01$). In this group, hierarchical classification showed a strong connection between the weight of parasitic lesions and levels of IL-4, IL-5, and IFN-$\gamma$, which belonged to the same cluster. In IL-33$^{-/-}$ mice, also at 4 months pi, lesion weight was not correlated with cytokine levels (Fig. 5B). IL-4, IL-5, IL-17A, IL-12p40, IFN-$\gamma$, IL-1$\beta$, and TNF-$\alpha$ were linked by significant pairwise correlation coefficients ($P$ between $\leq 0.05$ and $\leq 0.001$ for each possible combination) and were grouped together by low hierarchical clustering distances. In contrast, the WT group, IL-17F was significantly correlated with TNF-$\alpha$ ($P \leq 0.05$) and IFN-$\gamma$ ($P \leq 0.05$). Altogether, these results showed that IL-33$^{-/-}$ mice had a more efficient Th1/Th17 component at later stage of infection than WT mice. The systemic Th2 component was similar in both groups but was not associated with accelerated parasite growth rates in IL-33$^{-/-}$ mice, in contrast to

the WT group. This suggests that the mixed Th1/Th2 response observed in WT mice requires IL-33 to favor metacestode growth.

**IL-33 is expressed in the periparasitic liver tissue in human alveolar echinococcosis.** To assess whether the local role of IL-33 in the mouse peritoneal infection model could be extrapolated to human AE cases, we performed IL-33 quantification in protein extracts from parasitic, periparasitic, and liver parenchyma tissue biopsy specimens from three patients infected with *E. multilocularis* (Fig. 6A). We showed that IL-33 concentrations were higher in parasitic ($P \leq 0.05$) and periparasitic tissues (ns, not significant) compared to distant liver parenchyma tissues (Fig. 6B). Additionally, IL-33 and the soluble form of ST2 (ST2s) were quantified in the sera of patients suffering from AE, before and after radical surgery. As observed for mice, IL-33 was not detectable in sera, and ST2s was detected, but no significant differences were observed before and after surgery (Fig. 6C). We also performed immunofluorescence staining on liver biopsy specimens from five patients infected with *E. multilocularis* (the three previously mentioned patients and two additional ones, for whom a formalin-fixed paraffin-embedded biopsy specimen was available). In all samples, we observed an important neovasculature in the periparasitic granuloma, highlighted by the presence of numerous CD31$^+$ endothelial cells (Fig. 6D and Fig. S3). IL-33 was detected in the nuclei of endothelial cells (mainly in periparasitic tissues) and some hepatocytes (only in close parenchyma), but the density of IL-33$^+$ nuclei was similar between parenchyma and periparasitic granuloma (Fig. 6E). IL-33-expressing endothelial cells (CD31$^+$) were mainly located in the lymphoid infiltrate, an area where FoxP3$^+$ T$_{reg}$s were also highly prevalent (Fig. 6E).

## DISCUSSION

Alveolar echinococcosis is an atypical parasitic disease in that it shares more similarities with cancer than with other helminthic infections: it presents as a slowly and perpetually growing mass composed of parasitic vesicles, stromal and immune cells, and neovasculature and extracellular matrix, and it has the potential for metastatic dissemination (23). Following this analogy, we hypothesized that IL-33 favors AE, as it has been demonstrated in several tumor models that IL-33 is associated with a pro-tumorigenic microenvironment, especially through the accumulation of M2-like (CD206$^+$) tumor-associated macrophages (24–26), but also enhancing the activity of the T$_{reg}$s Th2 and ILC2 (27, 28). In our study, the lack of IL-33 led to a decrease of the parasitic growth during peritoneal mouse AE. The level of Th2 cytokines in mouse serum was correlated with the weight of the metacestodes in WT mice, in agreement with previous studies showing the role of these cytokines in parasite growth (5). However, this correlation was not observed in IL-33$^{-/-}$ mice, suggesting that systemic Th2 cytokines do not act directly on parasite growth but rather through one or several indirect mechanisms which require IL-33. Two of these could be the local activation of T$_{reg}$s and the polarization of macrophages. Indeed, IL-33, which was not detected in serum, was present in the periparasitic tissues of WT mice, where it was associated with an increase in M2-like macrophages together with the concentrations of the immunosuppressive cytokines IL-10 and TGF-$\beta$1, compared to that in IL-33$^{-/-}$ mice. Additionally, we observed that the proportions of CD206$^-$ and CD206$^+$ macrophages in peritoneal fluid were correlated with lesion weight in IL-33$^{-/-}$ but not in WT mice, supporting that the lack of IL-33 improved local response. In humans, IL-33 expression was increased in the parasitic tissue and localized in the nuclei of hepatocytes and endothelial cells of neovessels. This suggests that IL-33 could be released in the periparasitic microenvironment during human AE and interact with tolerogenic T$_{reg}$s and macrophages, as observed in mice.

CD206$^+$ macrophages have been described to promote angiogenesis and produce chemokines supporting tumor growth (29, 30). AE was shown to be associated with periparasitic accumulation of monocyte-derived M1 macrophages during the early phase of infection, which acquire a M2 phenotype (CD206$^+$) during the chronic stage of infection (31, 32). These macrophages are considered to favor parasite growth because they secrete the anti-inflammatory cytokine IL-10 and the fibrogenic cytokine TGF-$\beta$1, which has a dual role: slowing the parasite growth on one hand, and having immunosuppressive properties on the other (32–34). Interestingly, we observed in IL-33$^{-/-}$ mice both a decrease in CD206$^+$ tolerogenic macrophages and lower concentrations of IL-10 and TGF-$\beta$1 in the peritoneal cavity, compared to those in WT mice. We also observed a tendency towards lower concentrations of the Th1

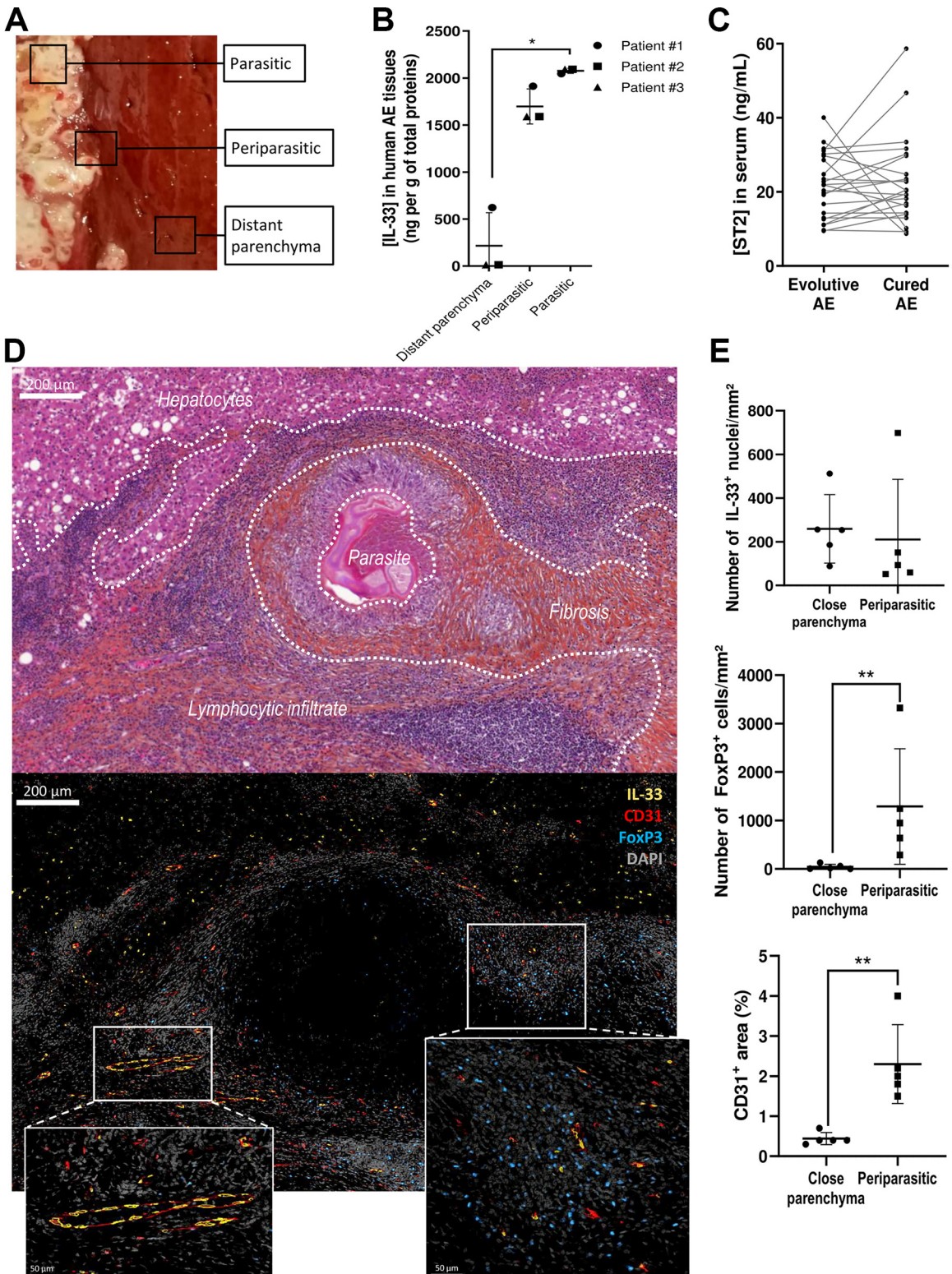

**FIG 6** IL-33 is detected in liver alveolar echinococcosis lesions from human patients. (A) Anatomical areas sampled for IL-33 quantification in the liver of patients with AE (*n* = 3). Squares show the location of biopsy samples. (B) IL-33 concentrations according to the sample site. Quantification after tissue lysis and enzyme-linked immunosorbent assay (ELISA) dosage. Friedman test and *post hoc* Dunn's multiple-comparison test (*, *P* ≤ 0.05). (C) Concentrations of soluble ST2 in sera of 18 patients infected with *E. multilocularis*, before and after curative surgery. Significance was tested using Wilcoxon matched-pairs signed-rank test. (D) Annotated hematoxylin and eosin staining and immunofluorescence staining of consecutive histological sections, using DAPI (4′,6-diamidino-2-phenylindole), anti-IL-33, anti-CD31, and anti-FoxP3 antibodies, with higher magnifications of structures of interest. Representative picture of biopsy sample slides obtained from 5 patients. (E) Quantification of density of IL-33⁺ nuclei (top), density of FoxP3⁺ cells (middle), and proportion of CD31⁺ area (bottom), in liver parenchyma and periparasitic granuloma, from immunofluorescence staining of human biopsy sample slides (*n* = 5). Mann-Whitney U test: **, *P* ≤ 0.01.

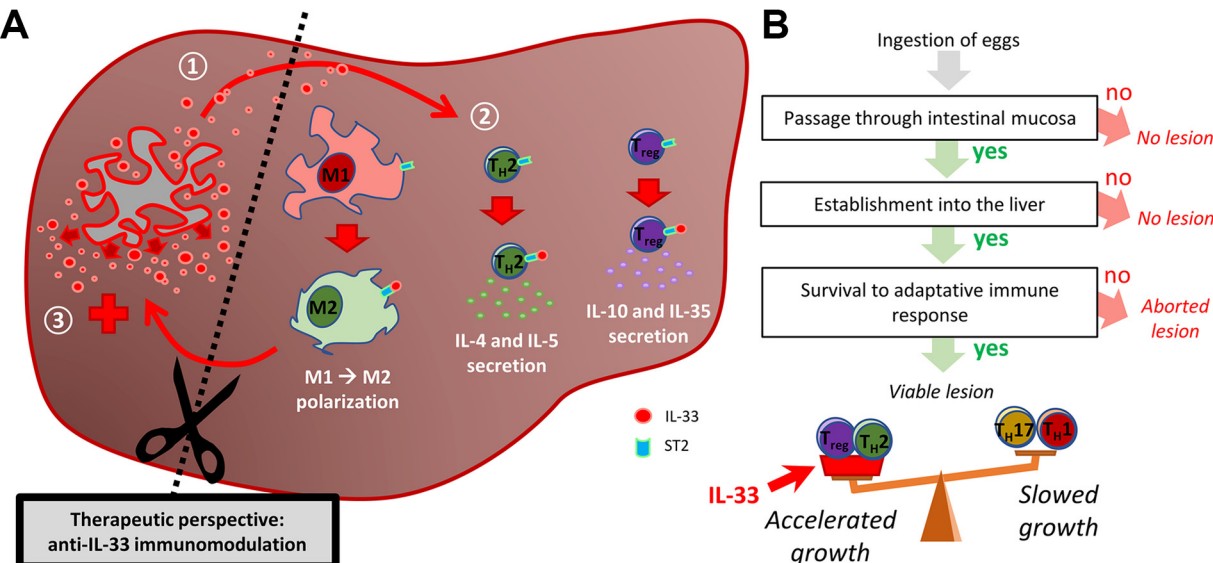

**FIG 7** Modulation of the host immune response by IL-33 and its impact on parasite growth. (A) Schematic presentation of the putative mechanism of action of IL-33 during alveolar echinococcosis. Step 1: IL-33 is released from neovasculature upon tissue damage by parasite growth. Step 2: IL-33 locally activates ST2$^+$ leukocytes, such as Th2$_{RM}$, ILC2s, and T$_{reg}$s, and contributes to polarization of macrophages into the "M2" phenotype. Step 3: ST2$^+$ leukocytes favor parasite growth by orientating immunity toward a tolerogenic phenotype. Blockade of IL-33 for a therapeutic purpose could limit the progression of the disease. (B) Schematic modeling of AE pathophysiology. After a host is contaminated with *E. multilocularis* eggs, the parasite must pass through 3 immunological events (framed text) to develop a viable metacestode. Once this is finished, immunological factors only impact the parasite's growth rate, with Th1/Th17 and Th2/T$_{reg}$ responses decreasing and increasing it, respectively. We showed that the alarmin IL-33 locally contributes to Th2 and T$_{reg}$ activity during late-stage AE, thus accelerating the course of the disease.

cytokines TNF-$\alpha$ (together with repression of *Tnfa* mRNA expression) and IL-2 and the Th1 chemokine CXCL-1, and higher concentrations of the Th2 chemokine CCL22, in IL-33$^{-/-}$ mice, which may seem contradictory with the decrease in M2 macrophages. However, it has been shown in a mouse model of airway injury that ST2 stimulation during M2 macrophage polarization induces *Tnfa*, *Il2*, and *Cxcl1* expression, and represses *Ccl22* expression, which strikingly correlates with our observations (Dagher et al. [35], supplemental data). Altogether, this suggests that during AE, endogenous IL-33 contributes to local M2 polarization and to the mixed Th1/Th2 local cytokinic profile.

IL-33 is also known to enhance the stability and immunosuppressive activity of T$_{reg}$s (36, 37). Using an *in vivo* mouse model of colonic inflammation, Schiering et al. (36) showed that upon tissue damage, IL-33 acts locally to stabilize T$_{reg}$s through ST2 signaling. T$_{reg}$s are major actors in AE pathophysiology (9), especially due to their production of IL-10 and TGF-$\beta$1 (8, 38), and it was shown that their depletion leads to smaller lesions (8). In our model, T$_{reg}$s were detected in the peritoneal cavity, suggesting that they can respond to the IL-33 released upon tissue damage by the parasite, as previously reported with other diseases (12, 25, 36). We did not observe increased proportions of T$_{reg}$s in WT mice compared to IL-33$^{-/-}$ mice, but it was shown previously that recruitment of ST2$^+$ T$_{reg}$s is independent of IL-33 signal, leading to similar or even higher proportions of T$_{reg}$s in IL-33$^{-/-}$ mice (39).

Altogether, our data suggest that IL-33 is involved in the tolerogenic microenvironment in the periparasitic tissues in late-stage AE. Immunofluorescence data from human samples suggest that the main source of the alarmin are the endothelial cells of the neovessels, but a role for the IL-33$^+$ periparasitic hepatocytes cannot be excluded. It should be noted that strong neoangiogenesis is characteristic of AE and the metacestode modulates the host response to favor it (40–42), leading to the accumulation of periparasitic CD31$^+$ IL-33$^+$ endothelial cells (41, 43). As the parasite grows, it induces damages to the neovessels, thus likely leading to the extracellular release of IL-33, which could be further cleaved into shorter and more active forms by mast cells or neutrophil proteases (44, 45). Following its release, IL-33 activates locally recruited ST2$^+$ cells, such as T$_{reg}$ and non-T$_{reg}$ CD4$^+$ cells (likely Th2 cells), and polarizes macrophages into M2 macrophages, which could increase Th2 and T$_{reg}$

responses and repress Th1 and Th17 responses (Fig. 7A). This microenvironment could favor parasite growth, leading to vascular damage, which in turn would release more IL-33. This vicious circle could be the target of new therapeutic strategies, combining albendazole medication with IL-33-blocking antibodies, which are currently in clinical trials for chronic inflammatory lung diseases (46).

One limitation of our study is the model of secondary peritoneal infection. In the natural cycle, rodents become infected with *E. multilocularis* by ingesting eggs, leading primarily to hepatic lesions. In this work, we inoculated metacestodes directly in the peritoneum of mice, using a different infective stage and leading to a different location of infection. However, this model, which is frequently used to study AE pathophysiology (9, 47), avoids the first pathophysiological uncertainty, i.e., the passage of the oncospheres through the intestinal mucosa, which conditions the establishment into the liver, thus permitting us to focus on the immunological response against the metacestode itself (Fig. 7B). Thanks to this model, we showed that local IL-33 is needed for modulation of the immune response to favor parasite growth. Additionally, the intraperitoneal AE model mimics advanced stages of AE, where dissemination in the peritoneal cavity is common (48).

## MATERIALS AND METHODS

**Ethics statement.** All experimental protocols on animals were conducted in compliance with French laws and the institution's guidelines for animal welfare (authors were authorized to conduct animal experimentation by "La Direction des Services Vétérinaires;" the project was authorized by the "Comité Régional d'Ethique d'Expérimentation Animal" license given by the "Ministère de l'Education Nationale et de la Recherche" [no. 15809-2018062910294494 and no. 28586-202011261535409]). Human sera were collected from 6 AE patients managed at the Rennes University Hospital (Rennes, France), with the approval of the local ethics committee (approval no 20.133), and from 12 AE patients managed at the Institute of Infectious Diseases of Bern (Switzerland) with the approval of the Cantonal Ethics Committee (382-15). Human liver samples were collected from 5 AE patients managed at the Rennes University Hospital with the approval of the local ethics committee (approval no 20.133).

**Animals and infection procedure.** The IL-33$^{-/-}$ strain of C57BL/6J mice was developed at the University of Toulouse (49). Animals were housed in individual cages and bred in specific pathogen-free conditions in a conventional animal facility (Arche Platform, Biosit, Rennes) with a 12 h dark-light cycle, under 25°C, 50% relative humidity, and HEPA-filtered air (Forma Scientific, Marietta, OH), with food and water *ad libitum*. IL-33$^{-/-}$ mice were crossed with C57BL/6J mice (Charles River Labs, Saint-Germain-Nuelles, France). Heterozygous mice were crossed in order to obtain IL-33$^{-/-}$ and WT littermate newborns. Mice were 8- to 10-weeks-old before peritoneal inoculation with 100 $\mu$L of metacestode suspension. Briefly, a clinical isolate of *E. multilocularis* (H95) (50) was maintained *in vivo* by serial peritoneal infections of BALB/c mice (Janvier Laboratories, Le Genest-Saint-Isle, France). Before the experiments, metacestodes were collected from an infected mouse, crushed, and sieved to remove debris. After an overnight sedimentation in phosphate-buffered saline (PBS) at 4°C, the pellet was washed with PBS and centrifuged for 5 min at 500 $\times$ *g* at 4°C. The pellet, which contained microvesicles and germinal cells, was suspended in PBS before peritoneal inoculation. Additionally, non-infected WT and IL-33$^{-/-}$ littermates were grouped together to constitute the "uninfected" group because they have similar phenotypes in the absence of inflammation (49, 51, 52). Submandibular blood samples were taken at different time points during the course of infection. At 4 months pi, mice were euthanized by cervical dislocation and blood, peritoneal lavages with 10 mL of PBS, parasitic lesions, and livers were collected.

**Protein extraction.** Total proteins were extracted from mouse and human tissues by lysis in RIPA buffer (50 mM Tris-HCl [pH 7.4], 1% Triton X-100, 150 mM NaCl, 10.7 mM magnesium chloride, 0.2% sodium deoxycholate, and 0.2% sodium dodecyl sulfate; all reagents provided by Sigma-Aldrich, Saint-Quentin-Fallavier, France) supplemented with phosphatase inhibitor (PhosSTOP EASYpack, Roche Diagnostics, Meylan, France) and protease inhibitor (cOmplete Mini, Roche Diagnostics, Meylan, France) cocktails, and crushed using the TissueLyser LT device (Qiagen, Les Ulis, France) at 50 Hz for 2 min. Supernatants were collected after centrifugation at 14,000 $\times$ *g* for 15 min at 4°C and sonicated. Before further use, protein concentration was determined by the Bradford method using the Bio-Rad Protein Assay Dye Reagent Concentrate (Bio-Rad, Marnes-la-Coquette, France).

**IL-33 and ST2 dosage.** Murine IL-33 was quantified in the sera and periparasitic tissues of mice using the DuoSet Mouse IL-33 assay (R&D Systems, Bio-Techne, Lille, France). Protein extracts from tissues were diluted to 0.5 g/L of total protein before assay. Human IL-33 and ST2s were quantified in serum and/or liver tissues using the Quantikine Human IL-33 and Quantikine Human ST2/IL-33R assays (R&D Systems), following the manufacturer's recommendations.

**Flow cytometry.** Flow cytometry was performed on peritoneal lavages and liver-infiltrating cells. Infiltrating cells were isolated from livers as previously described (53). Briefly, livers were crushed on 70-$\mu$m cell strainers and cells were sedimented for 1 h and fractioned using a 35% Percoll (Percoll Density Gradient Medium; GE Healthcare, Life Sciences, Thermo Fisher Scientific, Illkirch, France) gradient centrifugation. Red blood cells were then lysed using the erythrocyte-lysing reagent Easylyse (Agilent Technologies, Dako, Les Ulis, France). Two different panels of antibodies, one for lymphoid cells and another for myeloid cells (panels 1 and 2, respectively, Table S1), were used for staining. Cell suspensions were labeled for 30 min with LIVE/DEAD fixable

stain (Invitrogen, Thermo Fisher Scientific) for panel 1, or Zombie NIR Fixable Viability kit (Biolegend, Paris, France) for panel 2, to exclude dead cells. The FcR Blocking reagent (Purified NA/LE rat anti-mouse CD16/CD32, BD Biosciences, Le Pont de Claix, France) was used with both panels to avoid unspecific staining. Acquisition was performed with the LSR X-20 Fortessa flow cytometer (Becton, Dickinson, Plateform CytomeTRI, SFR Biosit, UMS CNRS 3480 INSERM 018, Rennes, France) and data were analyzed with Flowlogic v7.3 software (Inivai Technologies, Mentone Victoria, Australia). The gating strategy used is detailed in Fig. S1. Proportions of immune cell populations were determined by reporting the number of events out of the sum of all populations events.

**RNA extraction and cDNA synthesis.** Approximately 30 mg of tissues, either liver sample or periparasitic tissues (parasitic membranes and host conjunctive tissues), were homogenized using the TissueLyser LT device (Qiagen, Les Ulis, France) at 50 Hz for 2 min. Total RNAs were then extracted using the NucleoSpin RNA kit (Macherey-Nagel, Hoerdt, France), following manufacturer's recommendations. Complementary DNAs (cDNAs) were synthesized using the High-Capacity cDNA Reverse Transcription kit (Applied Biosystems, Thermo Fisher Scientific, Illkirch, France) and 2 $\mu$g of total RNA extract per sample. The obtained cDNAs were then diluted at 1:3 before amplification.

**mRNA expression quantification.** Quantification of mRNA expression was performed by qPCR using the CFX Opus 384 Real-Time PCR system (Bio-Rad, Marnes-la-Coquette, France). Each reaction was done in a 10-$\mu$L total volume, with the Power SYBR Green PCR Master Mix (Applied Biosystems, Thermo Fisher Scientific, Illkirch, France), 0.6 $\mu$M of each primer, and 2 $\mu$L of diluted cDNA. The primers used are detailed in Table S2 (52, 54–57). After an initial denaturation of 10 min at 95°C, 40 cycles of 95°C for 15 sec and then 60°C during 1 min were applied, followed by a melt curve from 60°C to 95°C. Results were expressed as the fold change expression of IL-33$^{-/-}$ mice compared to WT mice, using the threshold cycle ($2^{-\Delta\Delta CT}$) method. Mouse housekeeping genes were *Actb* and *Gapdh* for liver samples and *Gapdh* and *Ppid* for periparasitic tissues, and parasite housekeeping gene was *ActII*.

**Multiplex cytokine dosage.** Multiplex cytokine dosage in sera and peritoneal lavages of mice was performed by flow cytometry using Legendplex assays (BioLegend, Paris, France). Two kits with pre-mixed beads were used: the 13-plex Mouse macrophage/microglia panel and the 12-plex Mouse Th cytokine panel (BioLegend), following the manufacturer's recommendations. Prior to multiplex dosage, 3.5 mL of peritoneal lavage was centrifuged 40 min at 3,000 $\times$ $g$ using Amicon Ultra-15 10K centrifugal filter devices (Merck Millipore, Molsheim, France) for concentration of cytokines. Acquisition was done with the LSR X-20 Fortessa flow cytometer (Becton, Dickinson, Plateform CytomeTRI, SFR Biosit, UMS CNRS 3480, INSERM 018, Rennes, France).

**Immunofluorescence staining.** All reagents and devices were provided by Ventana Medical Systems (Tucson, AZ) unless stated otherwise. Paraffin-embedded tissue was cut at 4 $\mu$m, mounted on slides, and dried at 58°C for 12 h. Two consecutive cuts were processed: one was stained with hematoxylin and eosin, and immunofluorescent staining was performed on the other using the DISCOVERY ULTRA stainer. Following deparaffination with Ventana DISCOVERY Wash solution at 75°C for 8 min, antigen retrieval was performed by using Tris-based buffer solution CC1 at 95°C to 100°C for 40 min. Endogen peroxidase was blocked with Disc inhibitor for 12 min at 37°C. After rinsing with reaction buffer, slides were incubated at 37°C for 60 min with a 1:100 dilution of mouse anti-IL-33 (clone Nessy-1) primary antibody (Enzo Life Sciences, Villeurbanne, France). After rinsing, signal enhancement was performed using secondary antibody (anti-mouse-HRP) incubated for 16 min and incubation with the DISCOVERY Rhodamine kit for 20 min. Following denaturation with Ventana solution CC2 at 100°C for 8 min, and Disc inhibitor for 8 min at 37°C, slides were incubated at 37°C for 60 min with a 1:100 dilution of mouse anti-CD31 (clone JC70A) primary antibody (Dako, Agilent Technologies, Santa Clara, CA). After rinsing, signal enhancement was performed by incubation with the secondary antibody (anti-mouse-HRP) for 16 min and the DISCOVERY DCC kit for 20 min. Following denaturation with Ventana solution CC2 at 100°C for 8 min, and Disc inhibitor for 8 min at 37°C, slides were incubated at 37°C for 60 min with a 1:100 dilution of rabbit anti-FoxP3 (clone SP97) primary antibody (Abcam, Paris, France). After rinsing, signal enhancement was performed by incubation with secondary antibody (anti-rabbit-HRP) for 16 min and the Opal Polaris 780 Reagent Pack (Akoya Biosciences, Marlborough, MA) for 20 min. Slides were then counterstained for 4 min with DAPI (4',6-diamidino-2-phenylindole) and rinsed. After removal from the instrument, slides were manually rinsed, coverslipped, and digitized using the Pannoramic Confocal device (3DHISTECH, Budapest, Hungary). Digital slides were then read with the CaseViewer v2.4 software (3DHISTECH). Quantification of immunofluorescence staining was performed using QuPath v0.3.2 (58). IL-33$^+$ nuclei and FoxP3$^+$ cells were counted using an automated cell detection tool, and the proportion of CD31$^+$ area was quantified using a pixel classifier.

**Statistical analyses.** Data were analyzed with the GraphPad Prism v8.0.2 software (GraphPad, La Jolla, CA). All tests were two-tailed and a 5% $\alpha$ risk was applied. Differences between experimental groups were assessed using the Mann-Whitney U test (comparison between 2 groups) or the Kruskal-Wallis test with Dunn's multiple-comparison *post hoc* analysis (comparison between more than 2 groups). Evolution of mouse weight over time was assessed using a two-way analysis of variance (ANOVA) for repeated measures with Bonferroni's post-tests. Concentrations of cytokines in mouse sera were compared (i) within groups, to the baseline start (day 0), using the Kruskal-Wallis test with Dunn's multiple-comparison *post hoc* analysis and (ii) between groups using the Mann-Whitney U test. Slopes of the linear regression between the lesion weight and the log-transformed proportions of immune cells were tested using the F test. Pairwise correlation coefficients were calculated using R v4.0.3 (R Foundation for Statistical Computing, Vienna, Austria), after normalization of data by logarithmic transformation, and plotted as heatmaps using the heatmaply package (59). Concentrations of IL-33 in human tissues were compared using the Friedman test with Dunn's multiple-comparison *post hoc* analysis. Before- and after-surgery dosages of ST2 in human sera were compared using paired *t* tests. Immunofluorescence data were compared using the Mann-Whitney U test. Unless stated otherwise,

mouse infection data originated from a combination of two independent experiments. All data and statistical test results are provided in the supplemental file 1.

## SUPPLEMENTAL MATERIAL

Supplemental material is available online only.

**SUPPLEMENTAL FILE 1**, XLSX file, 0.1 MB.

**SUPPLEMENTAL FILE 2**, PDF file, 2 MB.

## ACKNOWLEDGMENTS

We thank the whole IRSET Team 2, especially Jacques Le Seyec and Claire Piquet-Pellorce, for their contributions. We thank the SFR Biosit, UMS CNRS, 3480-INSERM 018, Rennes (https://biosit.univ-rennes1.fr/) H2P2 platform, especially Pascale Bellaud, for their technical assistance with immunofluorescence stainings; the CytomeTRI platform; and the Arche animal house platform.

B.L.-S. was supported by the SNSF Project 310030_192072 fund.

B.A.: methodology, resources, investigation, data curation, formal analysis, validation, writing – original draft, visualization. C.M.: investigation, data curation, formal analysis. B.L.-S.: conceptualization, resources, writing – review & editing. J.-P.Gi.: resources, writing – review & editing. B.G.: conceptualization, writing – review & editing. J.-P.Ga.: resources, funding acquisition. M.S.: resources, funding acquisition, project administration. F.R.-G. and S.D.: conceptualization, methodology, investigation, supervision, project administration, writing – review & editing. All authors reviewed and agreed with the submitted version of the manuscript.

We have no conflicts of interest to declare.

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
