## [Reviewer comments · Microbiology Spectrum]

Microbiology Spectrum

Endogenous IL-33 accelerates metacestode growth during late-stage alveolar echinococcosis

Brice Autier, Christelle Manuel, Britta Lundström-Stadelmann, Jean-Philippe Girard, Bruno Gottstein, Jean-Pierre Gangneux, Michel Samson, Florence Robert-Gangneux, and Sarah DION

Corresponding Author(s): Sarah DION, Universite de Rennes 1

Review Timeline:

Submission Date:	October 18, 2022
Editorial Decision:	January 6, 2023
Revision Received:	January 27, 2023
Accepted:	January 28, 2023

Editor: Michael Ginger

Reviewer(s): Disclosure of reviewer identity is with reference to reviewer comments included in decision letter(s). The following individuals involved in review of your submission have agreed to reveal their identity: Henry McSorley (Reviewer #1)

Transaction Report:

DOI: <https://doi.org/10.1128/spectrum.04239-22>

January 6, 2023

Dr. Sarah DION
Université Rennes 1
2 Avenue Léon Bernard
Rennes, --- Select One --- 35043
France

Re: Spectrum04239-22 (Endogenous IL-33 accelerates metacystode growth during late-stage alveolar echinococcosis)

Dear Dr. Sarah DION:

Thank you for submitting your manuscript to Microbiology Spectrum, which has been reviewed by one expert reviewer (comments below). Thank you also for your patience whilst your study has been under review. I'm pleased to say the reviewer is supportive of your study but nonetheless a series of comments are offered for your consideration. In particular, concerns are raised around the ST2 flow cytometry staining where potentially additional controls need to be shown in order to convince all readers - suggestions for such controls are offered. Alternatively, your referee questions whether the ST2 data are absolutely necessary in order for you to convey the main interpretation.

Link Not Available

Sincerely,

Michael Ginger

Journals Department
Reviewer comments:

Reviewer #1 (Comments for the Author):

This manuscript describes the role of IL-33 in responses to echinococcus infection. Echinococcus is eliminated from the host through a TH1/TH17 response, while a Th2 response is ineffective against the parasite. The authors show that IL-33-deficient mice have reduced lesion sizes compared to WT mice, and this correlated with reduced IL-10 and increased IL-17F responses. In WT mice, lesion size correlated with Th2 immune responses, but this association was lost in IL-33-deficient mice. In human

echinococcus patients, IL-33 expression was increased in proximity to the parasite.

The study is generally well designed, and appropriately interpreted.

ST2 staining in Figure 2 is not at all convincing, it is very weak and I do not believe this is a true reflection of ST2 expression. Especially on e.g. CD4+ T cells, there should be only a sub-population of cells which would express ST2, whereas what is shown here (by MFI and histogram) is a shift of the entire population very slightly up the histogram: this would be more consistent with non-specific staining. The data should be analysed as % ST2 positive in each population. Also, there is no comparison here to uninfected controls. ST2 staining in uninfected controls must be shown in this figure. If a Th2 response is induced in WT mice, it would be expected that these Th2 cells would upregulate ST2, and therefore an increase in ST2 expression would be seen between WT uninfected and WT infected T cells. A population known to express high levels of ST2 (e.g. ILC2s, mast cells) should also be shown for comparison.

Where "uninfected" mice are shown, this is a pool of WT and IL-33-deficient mice (according to materials and methods). This approach is ok as long as the populations are similar in both genotypes of mice. It should be made clear also in the figure legends that this group consists of both WT and IL-33-deficient animals. The genotype of these controls should be made clear by e.g. using closed circles for WT and open circles for IL-33-deficient.

Authors should comment on the finding that although there is greater IL-33 expression in parasite-infected regions of human samples, there is no change in numbers of IL-33-positive cells. Therefore, presumably, the IL-33-producing cells must have greatly upregulated IL-33 expression?

Fig 2: Correlations show that there is a correlation with CD206- % of mono/macs but not CD206+ % of mono/macs. How can this be the case? As mono/macs must be, by definition, either CD206+ or CD206-, you would expect an inverse correlation with the CD206+ population.

Staff Comments:

Preparing Revision Guidelines

Please return the manuscript within 60 days; if you cannot complete the modification within this time period, please contact me. If you do not wish to modify the manuscript and prefer to submit it to another journal, please notify me of your decision immediately so that the manuscript may be formally withdrawn from consideration by Microbiology Spectrum.

This manuscript describes the role of IL-33 in responses to echinococcus infection. Echinococcus is eliminated from the host through a TH1/TH17 response, while a Th2 response is ineffective against the parasite. The authors show that IL-33-deficient mice have reduced lesion sizes compared to WT mice, and this correlated with reduced IL-10 and increased IL-17F responses. In WT mice, lesion size correlated with Th2 immune responses, but this association was lost in IL-33-deficient mice. In human echinococcus patients, IL-33 expression was increased in proximity to the parasite.

The study is generally well designed, and appropriately interpreted.

ST2 staining in Figure 2 is not at all convincing, it is very weak and I do not believe this is a true reflection of ST2 expression. Especially on e.g. CD4+ T cells, there should be only a sub-population of cells which would express ST2, whereas what is shown here (by MFI and histogram) is a shift of the entire population very slightly up the histogram: this would be more consistent with non-specific staining. The data should be analysed as % ST2 positive in each population. Also, there is no comparison here to uninfected controls. ST2 staining in uninfected controls must be shown in this figure. If a Th2 response is induced in WT mice, it would be expected that these Th2 cells would upregulate ST2, and therefore an increase in ST2 expression would be seen between WT uninfected and WT infected T cells. A population known to express high levels of ST2 (e.g. ILC2s, mast cells) should also be shown for comparison.

The authors thank reviewer 1 for this interesting point. The design of the cytometry panel did not allow to discriminate a population known to express high levels of ST2 such as ILC2s and mast cells. Additionally, these are probably rare in the peritoneal cavity of infected mice. However, the used anti-ST2 antibody was previously validated in another mouse experiment of the team, and showed a clear staining of a ST2⁺ subpopulation of Treg lymphocytes (see below, Figure from Noel G et al. Ablation of interaction between IL-33 and ST2⁺ regulatory T cells increases immune cell-mediated hepatitis and activated NK cell liver infiltration. *Am J Physiol Gastrointest Liver Physiol.* 2016 Aug 1;311(2):G313-23.).

Below, we provided dot plots from our study showing what we believe to be clear ST2⁺ subpopulations of T CD4⁺ cells. These were, as expected, increased in infected mice compared to uninfected controls for both CD4⁺FoxP3⁻ cells (2.1% vs 0.8%, respectively, p<0.05) and CD4⁺FoxP3⁺ cells (10% vs 4%, respectively, p<0.01).

For monocytes, macrophages and dendritic cells, no clear ST2⁺ subpopulation can be isolated. However, the ST2 receptor is lowly expressed on most ST2⁺ cell populations, and often homogeneously, resulting in a shift of the entire population. This has been described for macrophages (Dagher et al. IL-33-ST2 axis regulates myeloid cell differentiation and activation enabling effective club cell regeneration. *Nature Com* 2020; Yang et al. ST2/IL-33-Dependent Microglial Response Limits Acute Ischemic Brain Injury. *J Neuroscience* 2017.), or NK cells and ILCs (Kearley et al. Cigarette Smoke Silences Innate Lymphoid Cell Function and Facilitates an Exacerbated Type I Interleukin-33-Dependent Response to Infection. *Immunity* 2015). Except the FMO control we used in this study, the only way to prove a low expression of ST2 by an entire population would be to infect a group of ST2^{-/-} mice for comparison, what we cannot do in the due time.

We still believe in our statements but we cannot provide further data emphasizing the expression of ST2 on these cell populations. Since the reviewer feel it more appropriate, and because this does not call into question the rest of the manuscript, we deleted histograms and MFI analysis from Figure 2.

Where "uninfected" mice are shown, this is a pool of WT and IL-33-deficient mice (according to materials and methods). This approach is ok as long as the populations are similar in both genotypes of mice. It should be made clear also in the figure legends that this group consists of both WT and IL-33-deficient animals. The genotype of these controls should be made clear by e.g. using closed circles for WT and open circles for IL-33-deficient.

The authors agree with this suggestion. Figures 2-4 captions were modified in order to include the composition of the "uninfected" group. Also, WT and IL-33^{-/-} uninfected mice were showed as closed and open circles, respectively.

Authors should comment on the finding that although there is greater IL-33 expression in parasite-infected regions of human samples, there is no change in numbers of IL-33-positive cells. Therefore, presumably, the IL-33-producing cells must have greatly upregulated IL-33 expression?

IL-33 dosage by ELISA has been performed on 3 human biopsies, each sampled in 3 different areas. These areas were determined macroscopically and include parasitic tissue, border of the parasitic tissue and liver tissue.

Analysis of the number of IL-33⁺ nuclei has been made on histological sections of parasitic tissues. Thus, the IL-33⁺ hepatocytes ("close liver parenchyma") are adjacent to the

metacestode, and correspond to the macroscopic border of the parasitic lesion (or "periparasitic" in Fig 6A and 6B). In other words, the ELISA dosage in liver tissue does not correspond to the amount of IL-33 in the hepatic parenchyma adjacent to the parasite.

Fig 6A and 6B were modified: "Liver parenchyma" has been changed for "Distant parenchyma", to emphasize the difference with the "Close parenchyma" in Fig 6E.

Fig 2: Correlations show that there is a correlation with CD206⁻ % of mono/macs but not CD206⁺ % of mono/macs. How can this be the case? As mono/macs must be, by definition, either CD206⁺ or CD206⁻, you would expect an inverse correlation with the CD206⁺ population.

As shown in the initial Supp Fig 1, gating of CD206⁻ and CD206⁺ monocytes/macrophages did not include the totality of "monocytes/macrophages" events. Following reviewer's comment, gating was modified (monocytes-macrophages are now either CD206⁻ or CD206⁺) and new proportions of CD206⁺ macrophages were shown to be significantly correlated with the weight of the lesions, as expected. Fig 2B, 2D and manuscript were modified consequently.

January 28, 2023

Dr. Sarah DION
Universite de Rennes 1
2 Avenue Léon Bernard
Rennes, --- Select One --- 35043
France

Re: Spectrum04239-22R1 (Endogenous IL-33 accelerates metacestode growth during late-stage alveolar echinococcosis)

Dear Dr. Sarah DION:

Your manuscript has been accepted, and I am forwarding it to the ASM Journals Department for publication. You will be notified when your proofs are ready to be viewed.

Sincerely,

Michael Ginger
Editor, Microbiology Spectrum
